- 1 Feasibility of three-dimensional density tomography
- <sup>2</sup> using dozens of muon radiographies and Filtered

# **BackProjection for volcano**

- Shogo Nagahara<sup>1</sup>, Seigo Miyamoto<sup>1</sup>
- $\,$   $\,^1\!\mathrm{Earth}$  uake Research Institute, The University of Tokyo
- Correspondence to: Shogo Nagahara (<u>nagahara@eri.u-tokyo.ac.jp</u>)
- $\mathbf{7}$

Abstract. This study is the first trial to apply the method of filtered backprojection

(FBP) method to reconstruct three-dimensional (3D) bulk density images via cosmic-ray

muons, We also simulated three-dimensional reconstruction image with dozens of 12 muon radiographies using FBP method for a volcano and evaluated its practicality.

FBP method is widely used in X-ray and CT image reconstruction but has not been used in the field of muon radiography. One of the merits to use FBP method instead of

ordinary inversion method is that it doesn't require an initial model, while ordinaryinversion analysis need an initial model.

We also added new approximation factors by using data on mountain topography into existing formulas to successfully reduce systematic reconstruction errors. From a volcanic perspective, airborne radar is commonly used to measure and analyze mountain topography.

We tested the performance and applicability to the model of Omuroyama, a monogenetic scoria cone located in Shizuoka, Japan. As a result, it was revealed that the density difference between the original and reconstructed images depended on the number of observation points and the accidental error caused by muon statistics depended on the multiplication of total effective area and exposure period.

Combining above all things, we established how to evaluate an observation plan forvolcano using dozens of muon radiographies.

28

### 29 1 Introduction

### 30 1.1 Muon radiography and its principle

31Muon radiography is a method that can be used to make a map of the inner bulk 32density structures of large objects such as volcanoes, archeological targets, and so on, 33using secondary cosmic-ray muons. These muons are generated by the interactions 34between high energy primary cosmic rays (the main component is proton) and nuclei in 35the atmosphere. The flux, energy spectrum, and the zenith angle dependence of 36 secondary cosmic-ray muons have been well researched (e.g. Dorman, 2004; Honda et al., 2004; Patrignani et al., 2016; Nishiyama et al., 2016). Also their behavior 37 38 including energy loss in the various material have been investigated (Groom et al., 39 2001). Therefore, when we assume "density length", which is the integration of multiplication of density and material thickness, we can evaluate the number of 4041 penetrating muons. Muon detection technology also have been developed in the field of

particle physics and cosmic-ray physics. To make a bulk density map, we need to 4243measure not only the counts of penetrating muons from the target, but also the direction. For example, nuclear emulsion films (Morishima et al., 2017), hodoscope by 44scintillating plastic bars (Jourde et al., 2013), glass resistive plate chambers (Ambrosino 4546et al., 2015), multi-wire proportional chambers (Oláh et al., 2018) are capable to do that. 47By implementing these muon detectors around the target, we can get the penetrating 48muon flux for each direction from the detector, then by comparing to initial muon flux, we also get the attenuation of muons for each directions. By using the topographic data 49of the target, it is possible to lead the two-dimensional averaged bulk density from the 50muon attenuation and the path length of the target material. 5152The principle of X-ray radiography and muon radiography is very similar. There are 53two significant differences between these two methods: the first is the attenuation length. Typical X-ray beam can penetrate the material less than 1 meter water 5455equivalent. On the other hand, some muons can penetrate the order of kilo meter water 56equivalent because their kinetic energy is very high. The second difference is the origin of the source. The source of cosmic ray muons is completely environmental and we can't 57

control the flux while X-ray beam are generated by accelerating the electron artificially.
Typically, the number of observed muons is much smaller than ordinary X-ray
radiography.

The first significant result for volcanology was the two-dimensional bulk density 62 imaging of the shallow conduit in Mt. Asama by Tanaka et al., 2007a. Several 63 observation have been done after this research (e.g. Tanaka et al., 2007b; Lesparre et al., 64 2012; Tanaka et al., 2014).

### 66 1.2 Three-dimensional bulk density imaging

The internal structure of volcanoes gives important information for volcanology. For 68 example, the shape of shallow conduit affects the eruption dynamics (Ida, 2007). 69 However, muon radiography by only one direction makes just a 2D image, and this 70density is average of material along the muon path direction. Therefore, if we find some 71contrast in 2D density image, we can't distinguish the actual position of this density 72anomaly along muon path direction. To observe the real conduit shape, it is necessary to 73get the density image from different directions to reconstruct the three-dimensional 74bulk density image.

75Tanaka et al. (2010) attempted to observe the target from two directions in Mt. Asama. 76Nishiyama et al. (2014, 2017) conducted a 3D density analysis in Showa-Shinzan Lava 77Dome, combined with gravity observation data, which is also sensitive to density. 78Jourde et al. (2015) evaluated this joint-inversion method between muon radiography 79and gravity, and they observed and conducted 3D density analyses by using three-point 80 muon radiography and gravity data (Rosas-Carbajal et al., 2017). These previous 81 studies required prior information internal density distribution because of insufficient 82 observation data, and they were performed using inversion technique. 83 In this study, we propose the application of a 3D density-reconstruction analysis method using filtered back projection (FBP), which does not require prior information. This method is applied 84 85 to X-ray computed tomography (CT). However, muon radiography differs from X-ray CT in three 86 points. First, there is a constraint on the number of observation points and position. In X-ray CT, 87 there are hundreds of observation points, and each position is controllable. However, for muon 88 radiography, we can only use several dozen points, and the positions are limited because of 89 topography. Second, the cosmic-ray muon attenuation flux is not a simple exponential. Therefore, 90 the influence of muon statistical error depends on the results of 3D density, which is not trivial. Third, 91 in the case of muon radiography typically the amount of signal is much less than X-ray, because the 92source of cosmic-ray muon is completely environmental. Therefore, it is important to study the 93 features of FBP method in the case of realistic observations with various number of muon radiographies. So we should consider not only the reconstruction error by FBP method, but also how 94the error of muon statistics propagates to the final image. 95

- 96

#### 99 2 Method

The Radon transform is used to obtain projection images from all directions with 101 respect to a density distribution. In muon radiography, this corresponds to acquiring 102 observation data on density length from all directions. For three dimensions, the Radon 103 transform  $p(X, Z, \beta)$  of an object with density  $\rho(x, y, z)$  is given by the following:

$$p(X,Z,\beta) = \int \rho \left( -D\sin\beta + \frac{t}{\sqrt{1+X^2+Z^2}} (X\cos\beta + \sin\beta), D\cos\beta + \frac{t}{\sqrt{1+X^2+Z^2}} (X\sin\beta - 105\cos\beta), Z \right) dt,$$
(1)

- where x, y, and z are the positions in a 3D volume; X and Z are the tangents of 107 azimuth and elevation angle values, respectively;  $\beta$  is the observation point position at 108 a counterclockwise angle with respect to the y axis, and D is the distance between the 109 observation point and the origin. Figure 1 shows the geometric definition for these
- parameters.

Figure 1: A schematic of Radon Transform and the definition of parameters  $x, y, z, X, Z, \beta$ 115 and *D*.

In a 3D case, if observation data have an elevation angle and observation points only exist on the circumference, a complete inverse Radon transform does not exist. Therefore, approximation is needed. Feldkamp (1984) proposed one of the best methods to approximate a solution with a small elevation angle in two dimensions. This approximation is written as follows:

$$\rho(x, y, z) = \frac{1}{2} \int_0^{2\pi} d\beta \int_{-X_M}^{X_M} dX \frac{D}{L_2^2 \sqrt{1 + X^2 + Z^2}} p(X, Z_0, \beta) h(X_0 - X),$$
(2)

where  $Z_0 = z/(D - x \sin\beta - y \cos\beta)$ ,  $L_2 = \sqrt{1 + Z_0^2}(D + x \sin\beta - y \cos\beta)$ ,  $X_0 = (x \cos\beta + y \sin\beta)/L_2$ , and h(X) is a Ram-Lak filter (Ramachandran and Lakshminarayanan, 125 1971). A feature of this method is that it does not require the shape or initial model of 126 the object. However, when there is a density change in the vertical direction, the 127 accuracy of the approximation decreases. In many examples of volcanic muon

radiography, we obtain the shape of the volcano by using other methods; therefore, the 129 influence of changes in the shape can improve the accuracy of the approximation. To 130 estimate the elevation angle, we use the ratio of the path length of the observed muon 131  $q(X, Z_0, \beta)$  to the approximation of  $q_h(X, Z_0, \beta)$  (see Fig. 2), which can be written as 132 follows:

$$p'(X, Z, \beta) = \frac{q_h(X_m, Z, \beta_n)}{q(X_m, Z_{0n}, \beta_n)} p(X, Z, \beta),$$
 (3)

where  $p'(X, Z, \beta)$  is the approximation of the density length for the inverse Radon

transform. Finally, the reconstruction calculation formula can be written as follows: 136  $\rho(x, y, z)$

$$= \sum_{n=1}^{N} \delta\beta_n \sum_{m=1}^{M} \delta X_m \left( 1 - \frac{X_m}{D(\beta_n)} \delta D_n \right) \frac{D(\beta_n)}{L_2^2 \sqrt{1 + X_m^2}} \frac{p(X_m, Z_{0n}, \beta_n)}{q(X_m, Z_{0n}, \beta_n)} q_h(X_m, z, \beta_n) h(X_0 - X_m), (4)$$

where m, n is the index of X,  $\beta$ , respectively. We name this approximation "path length 139 normalization approximation (PLNA)."

Approximation path length  $q_h$ Reconstruction Point  $\rho(x, y, z)$ Length of this yellow line D'Muon path length qMuon path length qD Observation Point

Figure 2: Path length schematic and the approximation difference between Feldkamp 144 approximation and path length normalization approximation. In Feldkamp 145 approximation, the approximation density length is  $p' = D/D' \times p$ . In path length 146 normalization, the approximation density length is  $p' = q_h/q \times p$ .

### 150 3 Simulation

In this section, we describe the specific components of the simulation calculation. The

simulation calculation is divided into the following four steps:

| 153 | 1. Parameter setup                                            |
|-----|---------------------------------------------------------------|
| 154 | 2. Simulation calculation of the observed muon counts         |
| 155 | 3. Reconstruction calculation using data created in Step 2 $$ |
| 156 | 4. Calculations for evaluating the reconstruction results     |
| 157 |                                                               |
| 158 |                                                               |

#### 159 **3.1 Parameter setup for target and detector**

We simulated and reconstructed the density structure of Omuroyama, which is located in Shizuoka, Japan. We chose this volcano for two reasons. First, this volcano is easily observable from all directions because there are no large structures around the surrounding muon shields in a topographical view. Second, there are no occurrences of muon radiography for these large scoria hills. Omuroyama is a large scoria hill. We base the internal structural model of the large scoria hill on observations at the time of its formation (Luhr et al., 1993). However, there are currently no direct examples of these observations.

Figure 3 shows the contour map of the Omuroyama model used in the simulation.
We assume that the x axis is in the east-west direction, the y axis is in the northsouth direction, and the origin is the summit.

We configure the internal density distribution similar to a checkerboard with a side length of 100 m and a density of 1 and 2 g/cm<sup>3</sup>. We presume that the first internal density distribution is defined as the original image and is expressed as  $\rho^{ori}(x, y, z)$ .

The field of view was set to -2 to 2 (-63.4 to 63.4 in degrees) horizontally and 0 to 1 (0 to 45 in degrees) vertically, and the angular resolution was set to 0.04 (2.3 in degrees) in tangent. The observed muon statistics affect the density reconstruction error: the number of muons observed is proportional to the effective area of the device and the exposure period. The total effective area and exposure period *ST* of all muon devices was set as 1000 m<sup>2</sup> · days. For example, when the number of observation points is 16, each *ST* per point is 1000/16 =  $62.5m^2 \cdot days$ .

All observation points were assumed to be on the circumference of radius D = 500 m placed on the center (x, y) = (50 m, 50 m) of the mountain. The position of the observation points on the circumference is equal to the rotation angle from the reference line. The position  $\beta$ (rad) of the observation point is defined counterclockwise from the straight line parallel to the y axis and passes through the center (x, y) = (50 m, 50 m) of the mountain. The value of  $\beta$ , on which the observation point is placed, must always be one at  $\beta = 0$ , with the rest arranged at equal intervals along the circumference. For