# Peer review of "BackProjection for volcano"

_Geoscientific Instrumentation, Methods and Data Systems, 2018_

## Referee Comment (RC1) · Anonymous Referee #1 · 29 Jun 2018

This is a very nice paper although of limited applicability in real case Muography where only a few measurements can be made and at different altitudes.

The authors show that the Path Length Normalisation Approx. has greater precision with respect to the standard Feldkamp inversion, still it would have been perhaps more useful to focus on fewer observation points (3 to 8) and see whether an iterative computing intensive procedure would have produced better results.

Anyway, the results the athors obtain are significant and in my opinion the paper should

be published with minor revisions. See the following comments.

English needs revising, many instances of wrong grammar and unclear constructs. For example: 39) Therefore, when we assume "density length", which is the integration of multiplication of density and material thickness .... (integral ?)

40) Singular/plural mispelling (Muon detection technology also have been ...)

117) In a 3D case, if observation data have an elevation angle and observation points only exist on the circumference, a complete inverse Radon transform does not exist.

Typo or missing sentence: 215) Figure 4a shows the observation state at observation point A in figure 3, and Fig. 4b shows the theoretical muon count observation ðİŚĄ0(ðİŚŃ, ðİŚ■, 0) at that time. ðİŚĄ0(ðİŚŃ, ðİŚ■, ðİŻ¡) is the It is not suitable to use muon flux table in the region of 10 meter water equivalent or

Other comments: 19) From a volcanic perspective, airborne radar is commonly used to measure and analyze mountain topography. Topography usually derives form satellite or airborne imaging, and if you really want precision, laser scans not radar mapping. Please clarify, and give sources.

45) These citations are swapped, Ambrosino is plastic scintillators ... (hodoscope by scintillating plastic bars (Jourde et al., 2013), glass resistive plate chambers (Ambrosino et al., 2015), )

---

## Referee Comment (RC2) · Anonymous Referee #2 · 19 Aug 2018

The paper is dedicated to 3D methods developments in the field on muography. There is a real interest in this methodological work for all small and portable detectors operations such as nuclear emulsions for instance. This is more questionable for real-time muons hodoscopes. The objectives are clearly stated and the results reasonably presented for a first methodological approach. Real muons data would have been appreciated if available. This would have been more breaking-through. Apart from esthetical corrections and minor typos, I have a more technical question concerning the role of the a priori inputs one has to bring in the method. It is said that a small amount of a

priori information is needed but this sounds not true when one really implements it. For instance to stabilize the inversion you need to use constraints on the solutions, which correspond to real a priori information. I would like the authors to comment on this and to add a discussion paragraph on this item if possible.
* * *

---

## Author Comment (AC1) · 5 Sep 2018

Thank you for reading carefully.

Referee Comments: English needs revising.

Author Response: We changed phrases at each sentence. 39) integration → integral 41) Muon detection technology also have been → The muon detector have been

Referee Comments: Typo or missing sentence.

[Figure]

Author Response: 217) "N_0 (X,Z,$\beta$) is the" is deleted.

Referee Comments: 19) From a volcanic perspective, airborne radar is commonly used to measure and analyze mountain topography. Topography usually derives form satellite or airborne imaging, and if you really want precision, laser scans not radar mapping. Please clarify, and give sources.

Author Response: "radar" is our mistake. "LIDAR" is true. Geospatial Information Authority of Japan (GSI) used this method when they made 5m DEM data. GSI Japanese page https://fgd.gsi.go.jp/download/ref_dem.html

Referee Comments: 45) These citations are swapped, Ambrosino is plastic scintillators ... (hodoscope by scintillating plastic bars (Jourde et al., 2013), glass resistive plate chambers (Ambrosino et al., 2015), )

Author Response: Ambrosino et al.(2015) used both plastic scintillators and glass resistive plate chambers, so we added Ambrosino et al.(2015) as the example of plastic scintillators. 45) hodoscope by scintillating plastic bars (Jourde et al., 2013, Ambrosino et al., 2015)

Thank you and best regards, Nagahara Miyamoto

---

## Author Comment (AC2) · 5 Sep 2018

First of all, thank you for your careful reading.

Referee Comments: I have a more technical question concerning the role of the a priori inputs one has to bring in the method. It is said that a small amount of a priori information is needed but this sounds not true when one really implements it. For instance, to stabilize the inversion you need to use constraints on the solutions, which correspond to real a priori information. I would like the authors to comment on this and

to add a discussion paragraph on this item if possible.

Author Response: As stated in line 84, this analysis method is "forward problem analysis", not "inverse problem analysis". This analysis method can give results without initial density model. All simulation results are obtained without using a priori information, except for mountain shape data. The analysis method used in this paper can be directly applied to the data when observation is performed with the same setup in the simulation. If this response is different from what you really want to mention about (actually we're afraid of that), please let us inform.

Thank you and best regards, Nagahara Miyamoto